# Acute kidney injury and adverse renal events in patients receiving SGLT2-inhibitors: A systematic review and meta-analysis

Jan Menne [ID] *[◉], Eva Dumann [ID][◉], Hermann Haller, Bernhard M. W. Schmidt

Department of Nephrology and Hypertension, Hannover Medical School, Hannover, Germany

◉ These authors contributed equally to this work.
* menne.jan@mh-hannover.de

## Abstract

### Background

Sodium-glucose cotransporter-2 inhibitors (SGLT2is) represent a new class of oral hypoglycemic agents used in the treatment of type 2 diabetes mellitus. They have a positive effect on the progression of chronic kidney disease, but there is a concern that they might cause acute kidney injury (AKI).

### Methods and findings

We conducted a systematic review and meta-analysis of the effect of SGLT2is on renal adverse events (AEs) in randomized controlled trials and controlled observational studies. PubMed, EMBASE, Cochrane library, and ClinicalTrials.gov were searched without date restriction until 27 September 2019. Data extraction was performed using a standardized data form, and any discrepancies were resolved by consensus. One hundred and twelve randomized trials ($n = 96,722$) and 4 observational studies with 5 cohorts ($n = 83,934$) with a minimum follow-up of 12 weeks that provided information on at least 1 adverse renal outcome (AKI, combined renal AE, or hypovolemia-related events) were included. In 30 trials, 410 serious AEs due to AKI were reported. SGLT2is reduced the odds of suffering AKI by 36% (odds ratio [OR] 0.64 [95% confidence interval (CI) 0.53–0.78], $p < 0.001$). A total of 1,089 AKI events of any severity (AEs and serious AEs [SAEs]) were published in 41 trials (OR 0.75 [95% CI 0.66–0.84], $p < 0.001$). Empagliflozin, dapagliflozin, and canagliflozin had a comparable benefit on the SAE and AE rate. AEs related to hypovolemia were more commonly reported in SGLT2i-treated patients (OR 1.20 [95% CI 1.10–1.31], $p < 0.001$). In the observational studies, 777 AKI events were reported. The odds of suffering AKI were reduced in patients receiving SGLT2is (OR 0.40 [95% CI 0.33–0.48], $p < 0.001$). Limitations of this study are the reliance on nonadjudicated safety endpoints, discrepant inclusion criteria and baseline hypoglycemic therapy between studies, inconsistent definitions of renal AEs and hypovolemia, varying follow-up times in different studies, and a lack of information on the severity of AKI (stages I–III).

**Data Availability Statement:** All relevant data are within the manuscript and its Supporting Information files.

**Funding:** The author(s) received no specific funding for this work.

**Competing interests:** I have read the journal's policy and the authors of this manuscript have the following competing interests: JM has received lecture fees from Boehringer Ingelheim and Astra Zeneca. HH has received lecture fees, advisory board Consultant fees, and Research funds from Boehringer Ingelheim and Astra Zeneca.

**Abbreviations:** ACE, angiotensin converting enzyme; ADA, American Diabetes Association; AE, adverse event; AKI, acute kidney injury; ARB, angiotensin receptor blocker; CI, confidence interval; EASD, European Society for the Study of Diabetes; eGFR, estimated Glomerular Filtration Rate; FAERS, FDA Adverse Event Reporting System; FDA, Food and Drug Administration; MedDRA, Medical Dictionary for Regulatory Activities; NCBI, National Center for Biotechnology Information; OR, odds ratio; PRISMA, Preferred Reporting Items for Systematic Reviews and Meta-Analyses; RASi, renin–angiotensin system inhibitor; RCT, randomized controlled trial; SAE, serious AE; SGLT2i, sodium-glucose cotransporter-2 inhibitor; VEGF, Vascular Endothelial Growth Factor.

## Conclusions

SGLT2is reduced the odds of suffering AKI with and without hospitalization in randomized trials and the real-world setting, despite the fact that more AEs related to hypovolemia are reported.

## Author summary

### Why was this study done?

- Sodium-glucose cotransporter-2 inhibitors (SGLT2is) are a class of drugs used to treat high blood sugar in diabetes. They work by blocking the reuptake of filtered glucose by the kidney and therefore increase the loss of sugar in the urine, which also leads to increased water loss.

- SLGT2is have been shown to have beneficial effects on diabetes control and heart and long-term kidney function.

- However, there is a concern that these drugs can cause acute kidney injury, meaning a significant decline in kidney function happening over a short period of time that may or may not be reversible.

### What did the authors do and find?

- We conducted a database search to identify studies reporting on adverse effects from SLGT2i use.

- We found 112 randomized trials. Forty-one of these reported on acute kidney injury in a total of 68,159 patients. Patients on SGLT2is had 25% lower odds for this adverse effect. In 5 observational (nonrandomized) cohorts reporting on 83,934 patients, the odds of acute kidney injury were 60% less in patients taking SGLT2is.

- Ninety-two randomized studies with 81,763 patients reported on hypovolemia (fluid depletion); this was found to be more likely in patients not taking SGLT2is, with 20% higher odds.

### What do these findings mean?

- We could not detect an increased risk of acute kidney injury in patients taking SGLT2is. Patients taking SGLT2is had lower odds of suffering an acute kidney injury than those who did not, despite the fact that these drugs increase fluid loss by the body.

- Our findings indicate that fear of causing acute kidney injury should not stop practitioners prescribing SGLT2is.

- However, our analysis had some drawbacks, such as inconsistent definitions being used in some of the studies, different patient characteristics in the included studies, studies

being carried out for different lengths of time, and missing details about the severity of acute kidney injury.

## Introduction

Over the last few years, sodium-glucose cotransporter-2 inhibitors (SGLT2is) have been introduced for the treatment of patients with diabetes mellitus. In a recent meta-analysis of the 3 cardiovascular outcome trials, it was shown that SGLT2is reduce major cardiovascular events and the risk for hospitalization because of heart failure [1,2]. Furthermore, they preserve kidney function and reduce the risk of progression of renal disease [1,3]. As a consequence, SGLT2is feature prominently in the new 2018 ADA (American Diabetes Association)/EASD (European Society for the Study of Diabetes) consensus report and are recommended for patients with kidney disease and heart failure [4]. However, there is concern that SGLT2i may lead to acute kidney injury (AKI) due to hypovolemia, excessive decline in transglomerular pressure, and induction of renal medullary hypoxic injury [5]. Based on postmarketing FAERS (Food and Drug Administration Adverse Events Reporting System) reports, in June 2016, the US Food and Drug Administration (FDA) issued a warning that SGLT2is (especially canagliflozin and dapagliflozin) might cause AKI [6,7]. A meta-analysis performed in 2017 indicated that dapagliflozin and canagliflozin might indeed increase the risk of adverse renal events, whereas empagliflozin might be beneficial [8]. Since then, several large, well-conducted randomized studies and observational studies reporting the incidence of AKI/acute renal failure have been published.

We performed a systematic review and meta-analysis to address the question of whether, in patients treated with SGLT2is (most, but not all, of whom had type 2 diabetes), there was a difference in the risk of AKI, volume depletion, and other adverse renal outcomes compared to patients taking placebo or other oral hypoglycemic agents. We looked at the following 3 primary renal adverse event (AE) categories with decreasing specificity but increasing incidence: 1) serious AE (SAE) due to AKI, 2) serious and nonserious AEs due to AKI, and 3) composite renal AEs. Because hypovolemia-induced prerenal AKI is one of the potential pathophysiological mechanisms, we also collected data on hypovolemia-related AE. Additionally, we analyzed the effect of SGLT2is on AKI in real-world observational cohort studies.

## Methods

### Data sources and searches

A search of PubMed, EMBASE, and the Cochrane library was performed (for search terms, see S1 Table). Additionally, we checked trial records submitted to ClinicalTrials.gov for additional reports on serious AKI/acute renal failure in any SGLT2i trial. There was no prospectively registered protocol; however, search terms, inclusion criteria, and data collection form were pre-specified in an analysis plan and remained unaltered during the data collection and analysis (see S2 Text). The initial search was performed on 12 January 2019 and rerun twice, on 4 April and 27 September 2019. No publication date and language restrictions were applied. Reference lists of selected articles and reviews on SGLT2 inhibition were hand-searched for additional studies (Fig 1; see S2 Table and S3 Table for information source and AE definition).

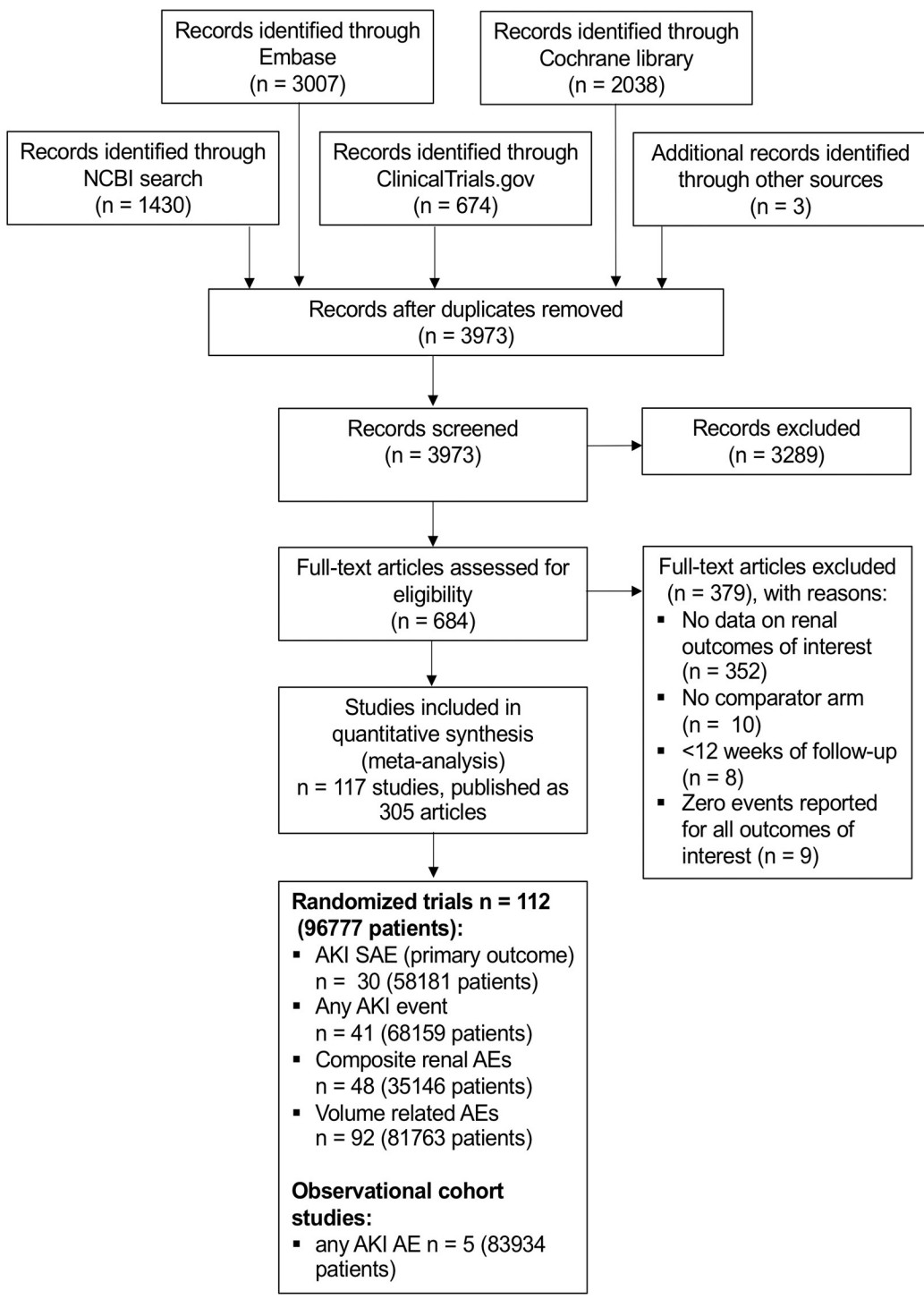

**Fig 1. PRISMA diagram.** AE, adverse event; AKI, acute kidney injury; NCBI, National Center for Biotechnology Information; PRISMA, Preferred Reporting Items for Systematic Reviews and Meta-Analyses; SAE, serious AE

## Study selection

Two investigators (JM, ED) independently reviewed all articles using a set of inclusion and exclusion criteria. In the case of differing results, a third investigator (BMWS) adjudicated. In

the first and second step, the articles were included or excluded based on their title and abstract. The remaining articles were then reviewed in full text. All articles that fulfilled the inclusion criteria were selected for data extraction (Fig 1).

We included all randomized controlled trials (RCTs) and cohort studies that reported AKI/ acute renal failure, a composite renal AE of interest, or volume depletion during treatment with SGLT2is and that had a control group, irrespective of the type of control (placebo or active control). After reviewing initial search results, a decision was made to exclude studies with a duration of <12 weeks and studies reporting zero events for all outcomes of interest. When several publications reported on the same study, we selected the record providing data for the longest follow-up period or the most complete set of data on outcomes of interest for further analysis. Observational studies were only included when they reported the incidence of AKI.

## Data extraction and quality assessment

Data were collected in duplicate by 2 investigators (JM, ED) independently using a standardized data extraction form. In the case of discrepancies, results were discussed by all authors, and consensus was established. The primary endpoint (AKI classified as SAE) was retrieved from the original published manuscripts and/or from the results published on ClinicalTrials. gov. In one case [2,3], uncertainty about the SAE event number in the placebo group because of a different reporting in the publications and the data provided on ClinicalTrials.gov was resolved by contacting the sponsor of the study. Secondary endpoints were "any AKI AE," "combined renal AEs," and "volume-depletion–related AEs."

For each study included in our analysis, we retrieved registration number, study type (RCT/observational), data source for the primary endpoint, drug received by the control group (placebo, no additional treatment, or alternative oral hypoglycemic agent), background therapy, blinding (yes/no), study duration, definition of secondary endpoints (any renal event and volume depletion), SGLT2i used, and dosage (mg) of SGLT2is. For the SGLTi group and control group in each study, respectively, we collected baseline characteristics (total number of patients, mean age, sex [% male], mean HbA1c at baseline, mean eGFR [estimated Glomerular Filtration Rate] at baseline, mean systolic blood pressure at baseline, coronary heart disease [%], eGFR <60 [%], AKI classified as SAE, AKI classified as AE or SAE) as well as numbers of events of interest in each category (AKI events reported as SAE or AE, any renal event, events related to volume depletion) (see S2 Table, S3 Table).

## Risk-of-bias assessment

Following the Cochrane risk-of-bias assessment, we classified each RCT contributing to the AKI endpoints as having low, high, or unclear risk based on the following criteria: random-sequence generation (selection bias), allocation concealment (selection bias), blinding (performance bias and detection bias), incomplete outcome data (attrition bias), and selective reporting (reporting bias) (see S1 Fig).

## Data synthesis and analysis

As principal summary measure, we calculated odds ratios (ORs) and 95% confidence intervals (CIs) for each study based on the event rate and patient number in each group. After the initial data extraction, it became evident that many studies reported separate results for different dose categories of the same SGLT2i. Metaregression analysis did not show a dose dependency for the occurrence of our primary endpoint (data not shown). Therefore, we pooled data from groups using different doses of SGLT2i within each study. A fixed-effects model was used for

combining results of studies. Studies with an event rate of 0 in both treatment and control arms (SAE AKI: 3 studies; any AKI AE: 2 studies; combined renal AE: 1 study; and volume-related AE: 6 studies) were excluded from the respective analysis. Cochrane's Q and $I^2$ were used for assessing heterogeneity between studies and a funnel plot to check for publication bias (S2 Fig). Following review of initial results, we carried out further non-prespecified sub-group analyses to compare different SGLT2is for the "combined renal AE" endpoint and "volume depletion" endpoints. For comparison of drugs, mixed-effects analysis was used, i.e., using a random-effects model for comparison within drugs and fixed effects for comparison across different drugs. All analyses were performed using the program Comprehensive Meta-Analysis (Version 2.2.064, Biostat, Englewood, NJ, USA). See S1 Text for further details.

## Results

We included a total of 112 RCTs with 96,777 patients and 5 observational cohorts with 83,934 patients in this meta-analysis. 41 RCTs with a total of 68,159 patients reported at least one AKI event. The risk of bias for the 41 RCTs with an AKI event is summarized as follows: 32 RCTs reported adequate random-sequence generation and adequate allocation concealment, and 34 reported adequate outcome data for SAE AKI, but only 14 for any AKI AE. All of the trials were funded by industry. More information is presented in S1 Fig.

### SAEs of AKI in RCTs

In 30 randomized trials with a total of 58,181 patients and a mean follow-up of 140 weeks, 410 SAEs due to AKI (384 with AKI, 25 with acute renal failure, and 1 with acute prerenal failure) were reported. In 33,234 SGLT2i-treated patients, 185 SAEs were observed, and in 24,947 control patients, 225 SAEs were observed (Fig 2). 23 trials compared SGLT2is against placebo and 7 trials against another oral hypoglycemic agent. SGLT2is diminished the odds of an SAE AKI by 36% (OR 0.64 [95% CI 0.53–0.78], $p < 0.001$). The effect size was comparable between dapagliflozin (OR 0.62 [95% CI 0.48–0.80], $p \leq 0.001$) and empagliflozin (OR 0.65 [95% CI 0.43–0.97], $p = 0.037$). The reported events were fewer in the canagliflozin studies, and the treatment effect showed a trend towards a reduction of SAE AKIs (OR 0.72 [95% CI 0.43–1.20], $p = 0.20$). For the other SGLT2is, low numbers of events are reported, so larger studies are required. No heterogeneity was detectable ($I^2 = 0$, Cochrane's Q $p = 0.99$). Removal of a single study did not affect the overall outcome (S3 Fig).

### AEs of AKI in RCTs

Seventeen trials with 41,095 patients and a mean follow-up of 173 weeks also reported acute renal failure events as part of a combined AKI endpoint, which included events classified as nonserious AEs (Fig 3, S1 Fig). The control patients received placebo in 12 trials, an alternative oral hypoglycemic agent in 4 trials, and subcutaneous semaglutide in 1 trial. In 23,035 SGLT2i-treated patients, 503 events of acute renal failure occurred, whereas in the control group, 461 AKI events occurred in 18,060 patients (OR 0.77 [95% CI 0.63–0.84], $p < 0.001$). The effect size was comparable between empagliflozin-, dapagliflozin-, and canagliflozin-treated patients ($p = 0.680$). No heterogeneity was detectable ($I^2 = 0$, Cochrane's Q $p = 0.85$).

Overall, 41 trials reported at least one SAE or a nonserious AKI AE (S4 Fig). In total, 1,089 events were reported in these trials. In the 38,441 SGLT2i-treated patients, there were 550 AKI events, compared to 539 events in 29,718 patients in the placebo arm. Overall, SGLT2is had a positive effect, with an odds reduction of 25% (OR 0.75 [95% CI 0.66–0.84], $p < 0.001$). No heterogeneity was detectable ($I^2 = 0$, Cochrane's Q $p = 0.98$).

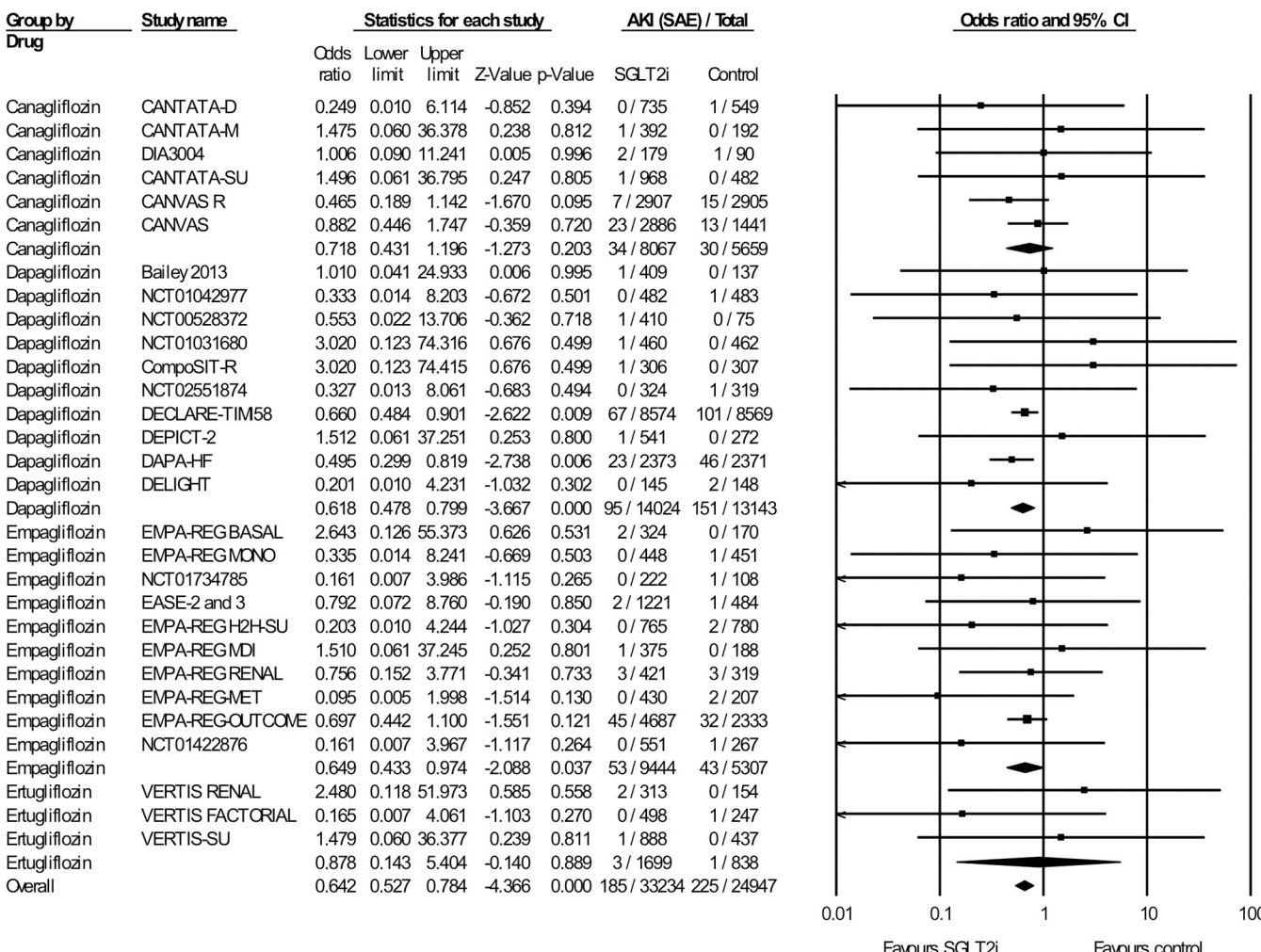

| Group by Drug | Study name | Statistics for each study | | | | | AKI (SAE) / Total | | Odds ratio and 95% CI |
|---|---|---|---|---|---|---|---|---|---|
| | | Odds ratio | Lower limit | Upper limit | Z-Value | p-Value | SGLT2i | Control | |
| Canagliflozin | CANTATA-D | 0.249 | 0.010 | 6.114 | -0.852 | 0.394 | 0 / 735 | 1 / 549 | |
| Canagliflozin | CANTATA-M | 1.475 | 0.060 | 36.378 | 0.238 | 0.812 | 1 / 392 | 0 / 192 | |
| Canagliflozin | DIA3004 | 1.006 | 0.090 | 11.241 | 0.005 | 0.996 | 2 / 179 | 1 / 90 | |
| Canagliflozin | CANTATA-SU | 1.496 | 0.061 | 36.795 | 0.247 | 0.805 | 1 / 968 | 0 / 482 | |
| Canagliflozin | CANVAS R | 0.465 | 0.189 | 1.142 | -1.670 | 0.095 | 7 / 2907 | 15 / 2905 | |
| Canagliflozin | CANVAS | 0.882 | 0.446 | 1.747 | -0.359 | 0.720 | 23 / 2886 | 13 / 1441 | |
| Canagliflozin | | 0.718 | 0.431 | 1.196 | -1.273 | 0.203 | 34 / 8067 | 30 / 5659 | |
| Dapagliflozin | Bailey 2013 | 1.010 | 0.041 | 24.933 | 0.006 | 0.995 | 1 / 409 | 0 / 137 | |
| Dapagliflozin | NCT01042977 | 0.333 | 0.014 | 8.203 | -0.672 | 0.501 | 0 / 482 | 1 / 483 | |
| Dapagliflozin | NCT00528372 | 0.553 | 0.022 | 13.706 | -0.362 | 0.718 | 1 / 410 | 0 / 75 | |
| Dapagliflozin | NCT01031680 | 3.020 | 0.123 | 74.316 | 0.676 | 0.499 | 1 / 460 | 0 / 462 | |
| Dapagliflozin | CompoSIT-R | 3.020 | 0.123 | 74.415 | 0.676 | 0.499 | 1 / 306 | 0 / 307 | |
| Dapagliflozin | NCT02551874 | 0.327 | 0.013 | 8.061 | -0.683 | 0.494 | 0 / 324 | 1 / 319 | |
| Dapagliflozin | DECLARE-TIMI58 | 0.660 | 0.484 | 0.901 | -2.622 | 0.009 | 67 / 8574 | 101 / 8569 | |
| Dapagliflozin | DEPICT-2 | 1.512 | 0.061 | 37.251 | 0.253 | 0.800 | 1 / 541 | 0 / 272 | |
| Dapagliflozin | DAPA-HF | 0.495 | 0.299 | 0.819 | -2.738 | 0.006 | 23 / 2373 | 46 / 2371 | |
| Dapagliflozin | DELIGHT | 0.201 | 0.010 | 4.231 | -1.032 | 0.302 | 0 / 145 | 2 / 148 | |
| Dapagliflozin | | 0.618 | 0.478 | 0.799 | -3.667 | 0.000 | 95 / 14024 | 151 / 13143 | |
| Empagliflozin | EMPA-REG BASAL | 2.643 | 0.126 | 55.373 | 0.626 | 0.531 | 2 / 324 | 0 / 170 | |
| Empagliflozin | EMPA-REG MONO | 0.335 | 0.014 | 8.241 | -0.669 | 0.503 | 0 / 448 | 1 / 451 | |
| Empagliflozin | NCT01734785 | 0.161 | 0.007 | 3.986 | -1.115 | 0.265 | 0 / 222 | 1 / 108 | |
| Empagliflozin | EASE-2 and 3 | 0.792 | 0.072 | 8.760 | -0.190 | 0.850 | 2 / 1221 | 1 / 484 | |
| Empagliflozin | EMPA-REG H2H-SU | 0.203 | 0.010 | 4.244 | -1.027 | 0.304 | 0 / 765 | 2 / 780 | |
| Empagliflozin | EMPA-REG MDI | 1.510 | 0.061 | 37.245 | 0.252 | 0.801 | 1 / 375 | 0 / 188 | |
| Empagliflozin | EMPA-REG RENAL | 0.756 | 0.152 | 3.771 | -0.341 | 0.733 | 3 / 421 | 3 / 319 | |
| Empagliflozin | EMPA-REG-MET | 0.095 | 0.005 | 1.998 | -1.514 | 0.130 | 0 / 430 | 2 / 207 | |
| Empagliflozin | EMPA-REG-OUTCOME | 0.697 | 0.442 | 1.100 | -1.551 | 0.121 | 45 / 4687 | 32 / 2333 | |
| Empagliflozin | NCT01422876 | 0.161 | 0.007 | 3.967 | -1.117 | 0.264 | 0 / 551 | 1 / 267 | |
| Empagliflozin | | 0.649 | 0.433 | 0.974 | -2.088 | 0.037 | 53 / 9444 | 43 / 5307 | |
| Ertugliflozin | VERTIS RENAL | 2.480 | 0.118 | 51.973 | 0.585 | 0.558 | 2 / 313 | 0 / 154 | |
| Ertugliflozin | VERTIS FACTORIAL | 0.165 | 0.007 | 4.061 | -1.103 | 0.270 | 0 / 498 | 1 / 247 | |
| Ertugliflozin | VERTIS-SU | 1.479 | 0.060 | 36.377 | 0.239 | 0.811 | 1 / 888 | 0 / 437 | |
| Ertugliflozin | | 0.878 | 0.143 | 5.404 | -0.140 | 0.889 | 3 / 1699 | 1 / 838 | |
| Overall | | 0.642 | 0.527 | 0.784 | -4.366 | 0.000 | 185 / 33234 | 225 / 24947 | |

0.01    0.1    1    10    100

Favours SGLT2i          Favours control

**Fig 2. Effect of SGLT2is on serious AKI AEs in RCTs (Ease 2 and 3 are two separate trials).** AE, adverse event; AKI, acute kidney injury; CI, confidence interval; RCT, randomized controlled trial; SAE, serious AE; SGLT2i, sodium-glucose cotransporter-2 inhibitor.

## AKI when eGFR < 60 ml/min

Four studies reported the AKI SAE rate in 4,983 patients with a baseline eGFR < 60 ml/min (S5 Fig). For 3,077 SGLT2i-treated patients, 48 SAEs were observed; there were 36 SAEs in 1,906 control patients. The difference was not significant, but the effect size was comparable to the effect size in the total cohort (OR 0.76 [95% CI 0.49–1.18], $p$ = 0.22).

## Renal composite AEs

A renal composite AE was reported in 48 studies, with a total of 35,146 patients and a mean follow-up of 92 weeks. Definitions of renal composite AEs varied between the different studies and included acute renal failure, renal impairment, and renal failure but also increase in creatinine or decline of eGFR, without differentiation between chronic and more acute changes in kidney function (S2 Table). It should be noted that only one of the studies on empagliflozin used this form of combined AE definition. 1,010 renal composite AEs occurred in 20,719 patients treated with a SGLT2i versus 809 events in 14,427 control patients (S6 Fig), with an OR of 0.98 [95% CI 0.89–1.09], $p$ = 0.74). Considerable heterogeneity was detectable ($I^2$ = 33.9,

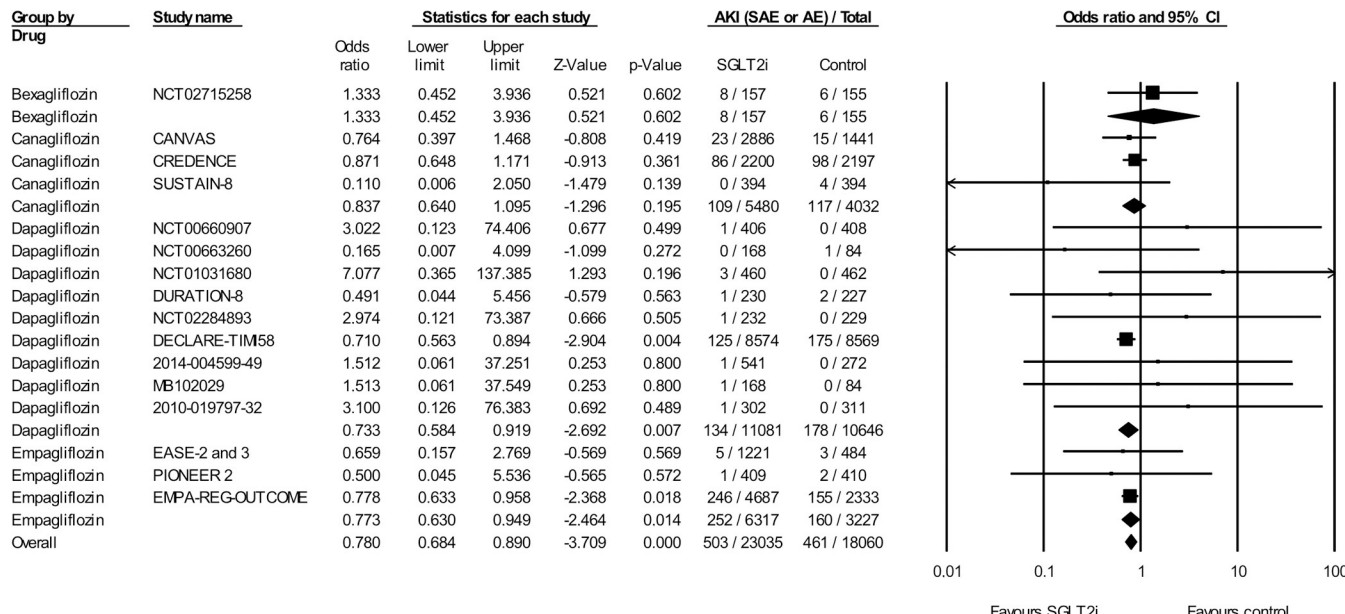

**Fig 3. Effect of SGLT2is on serious and nonserious AKI AEs in RCTs (Ease 2 and 3 are two separate trials).** AE, adverse event; AKI, acute kidney injury; CI, confidence interval; RCT, randomized controlled trial; SAE, serious AE; SGLT2i, sodium-glucose cotransporter-2 inhibitor.

Cochrane's Q $p$ = 0.013). Subgroup analysis did not show a difference caused by the drug used ($p$ = 0.722).

## Hypovolemia-related AEs

Volume-depletion–related AEs were reported in 92 studies with a total of 81,763 patients and a mean follow-up of 114 weeks. The outcome was defined inconsistently across different studies (S2 Table). 1,581 hypovolemia-related AEs occurred in 32,987 patients treated with an SGLT2i compared to 912 events in 48,776 patients taking in the control group (S7 Fig). The control group received placebo in 78 studies, an alternative oral hypoglycemic agent in 13 studies, and no additional treatment in 1 study. This AE was significantly more common in the SGLT2i-treated patients (OR 1.20 [95% CI 1.10–1.31], $p$ < 0.001). Empagliflozin (OR 1.10 [95% CI 0.91–1.33], $p$ = 0.30) and dapagliflozin (OR 1.10 [95% CI 0.96–1.25], $p$ = 0.29) did not increase the OR, but canagliflozin-treated patients had an OR of 1.39 [95% CI 1.18–1.65], $p$ < 0.001). However, comparison of effects across the different drugs showed no statistically significant difference of the estimates ($p$ = 0.41). Accordingly, no heterogeneity was detectable ($I^2$ = 0, Cochrane's Q $p$ = 0.546)

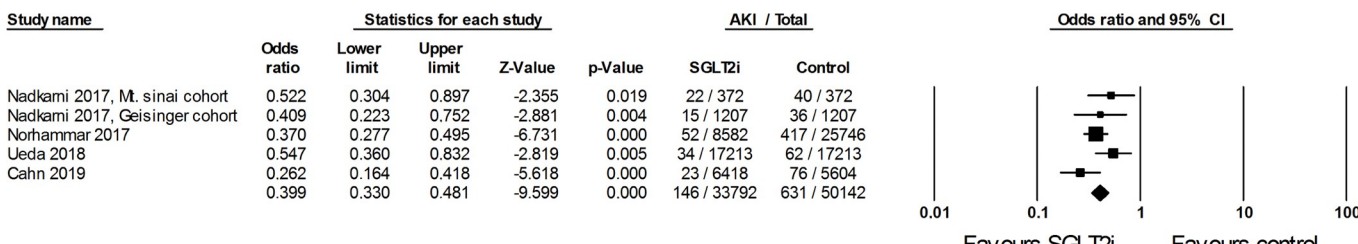

**Fig 4. Effect of SGLT2is on AKI in observational cohort studies.** AKI, acute kidney injury; CI, confidence interval; SGLT2i, sodium-glucose cotransporter-2 inhibitor.

## AKI in observational studies

The outcomes listed above were reported in RCTs. We also analyzed observational cohort studies if they reported on our main outcome of interest, AKI. This was the case for 5 observational cohort studies (2 of which were published in a single article). The definitions of AKI varied between the 5 cohorts (see S3 Table). A total of 83,934 patients with a follow-up between 24–65 weeks were analyzed. In 33,792 SGLT2i-treated patients, 148 patients developed an acute renal failure. In the control group, 631 out of 50,142 patients had an event. Overall, SGLT2is had a positive effect, with 60% lower odds of an adverse renal event (OR 0.40 [95% CI 0.33–0.48], $p < 0.001$) (Fig 4). Four of the studies used propensity score matching to adjust for baseline differences and reported an OR between 0.37 and 0.55. The study by Cahn and colleagues did not perform any matching [9]. We therefore repeated the analysis and excluded this trial. The OR increased slightly to OR 0.43 (95% CI 0.35–0.53), $p < 0.001$ (S8 Fig). No heterogeneity was detectable between the propensity-matched cohort studies ($I^2 = 0$, Cochrane's Q $p = 0.42$).

## Discussion

SGLT2is are important new compounds for the treatment of patients with diabetes, and there is a growing body of evidence supporting their beneficial effect in slowing eGFR decline and preventing progressive kidney damage. However, there is concern that they might cause AKI [5,7,10]. In this systematic review, >1,800 AKI events were identified in more than 150,000 patients in 41 RCTs and 5 observational cohorts. In the RCTs, SGLT2is reduced the odds of AKI requiring hospitalization by 36% and the odds of any form of an acute renal failure by 25%. Importantly, the effect was comparable for the different SGLT2is (canagliflozin, dapagliflozin, and empagliflozin). These results are supported by 4 observational propensity-score–matched cohorts in which the odds of AKI were reduced by 45%–64% when SGLT2is were prescribed. In one additional, recently published nonmatched cohort, the risk reduction was as high as 74% [9]. Because this systematic review is based on a high AKI event rate and we observed no heterogeneity, we conclude that SGLT2is may not only reduce the progression of chronic kidney disease but also have a preventive effect on AKI.

AKI is defined by an acute worsening of the kidney function and typically staged according to the increase in serum creatinine or reduction in urine output over 48 h to 7 days. Once widely believed to represent a benign and transient problem, it is now understood as a risk factor for an increased short-term and long-term mortality, chronic kidney disease, and end-stage renal disease. The severity of AKI (stages I–III) correlates with these outcome parameters. The evidence to date is summarized in several reviews [11–13]. In a large meta-analysis, roughly 20% of in-hospital patients experienced an AKI episode; likewise, a significant number of outpatients were affected [14]. It is estimated that every year, 13.3 million people will suffer from AKI. Several prediction models have been developed, and diabetes is the second most commonly used parameter beside age [15]. Therefore, a drug simultaneously improving glucose control and protecting against AKI would be advantageous.

To date, there is no mechanistic explanation why SGLT2is could prevent AKI despite the fact that a higher rate of hypovolemia, a well-known risk factor for acute prerenal failure, is reported. In 2 animal studies, administration of an SGLT2i attenuated ischemia-reperfusion injury and AKI [16,17]. The authors observed less tubular apoptosis and less peritubular vascular rarefication due to a higher local VEGF (Vascular Endothelial Growth Factor) production, with a protective effect against renal fibrosis. Interestingly, this positive effect was seen even though the animals were not diabetic. Because sodium transport is ATP-dependent, it has been postulated that a reduction in sodium reabsorption through SGLT2is in the S1 and S2

tubular segments leads to reduced tubular oxygen consumption [18]. Other SGLT2i-induced changes that could lead to tubular protection include increased erythropoietin production, suppression of peritubular inflammation and fibrosis, and increased use of ketone bodies as an alternative fuel source [18]. Biomarker studies in patients treated with dapagliflozin for 6 weeks showed a reduction of several tubular damage markers (kidney injury molecule-1, neu-trophil gelatinase-associated lipocalin, and liver-type fatty acid binding protein), suggesting that there may indeed be some tubular alteration that might decrease tubular susceptibility to AKI [19]. Further experimental studies to explore the underlying mechanisms are required.

In many RCTs, a combined renal AE endpoint was reported, and, in contrast to the effect on AKI, patients treated with an SGLT2i had no benefit. We believe that this discrepancy is due to the inconsistent criteria used to define the combined renal adverse outcome. Event defi-nitions did not differentiate between acute and chronic changes, and often a change in eGFR or creatinine was reported as an AE. However, SGLT2is lead to a reversible creatinine increase and decline of the eGFR due to their well-described effects on tubuloglomerular feedback, with constriction of the afferent arteriole as a result of increased sodium delivery to the macula densa at the distal tubule [3]. Therefore, such AE events are probably of little clinical relevance. Of note, >70% of patients in the large outcome trials received inhibitors of the renin–angio-tensin system (RASis), with similar proportions of patients treated with RASis in the SGLT2i and comparator groups (see S4 Table). Activation of RAS signaling causes efferent arteriolar constriction via angiotensin II; thus, RAS inhibition dilates the efferent arteriole and reduces filtration pressure. A combination of preglomerular constriction through SGLT2is and post-glomerular dilation under RASis would be expected to cause an increased risk of AKI. A recent subgroup analysis of data from the EMPA-REG OUTCOME trial by baseline background medications found a slightly increased risk of acute renal failure in patients on angiotensin converting enzyme (ACE) inhibitors or angiotensin receptor blockers (ARBs) compared to patients not taking these drugs [20]. However, empagliflozin was able to improve the risk of incident or worsening nephropathy regardless of ACE inhibitor/ARB use, and the risk of acute renal failure with ACE inhibitor/ARB use tended to be lower in patients also taking empagli-flozin (for patients on ACE inhibitors/ARBs, acute renal failure rates were 5.4% and 5.9% in empagliflozin-treated patients versus 7% in the placebo group).

AEs related to volume depletion were more commonly observed in SGLT2i-treated patients. For canagliflozin and sotagliflozin, the difference was significant against the control group. This was not the case for all other SGLT2is. However, in a mixed-effects analysis, no significant difference was observed between the compounds.

The study has 6 major limitations. 1) We used safety endpoints as reported in the corre-sponding publications. These endpoints are not validated by an endpoint committee. How-ever, the vast majority of randomized SGLT2i studies published so far used MedDRA (Medical Dictionary for Regulatory Activities) terminology to classify the AEs, leading to some stan-dardization across the different studies. To further enhance the value of the analysis, we selected an SAE of AKI as our primary outcome. SAEs are vigorously monitored, and a more detailed description of the event is collected. 2) Non-SAE AKI events were only reported in 17 studies, including the DECLARE, EMPAREG-OUTCOME, CANVAS, and CREDENCE stud-ies, and they are only mentioned as part of a broader composite renal AE safety endpoint in the rest of the studies. The interpretation of these results is difficult. 3) In some studies, there was probably an under-reporting of AKI events. The best example is the CANVAS study. It is not plausible that 0.9% of patients had an SAE due to AKI but only 1% had an AKI. This would suggest that only 0.1% had a nonserious AKI event. This is far below the event rate reported in the other outcome studies. 4) None of the included studies reported AKI stages. Because there is a clear link between the severity of AKI and short-term and long-term

consequences, these data would be very helpful in understanding the effect of SGLT2is on AKI. 5) While most RCTs compared the effect of SGLT2is against placebo, a number of studies had an active comparator group receiving an alternative oral hypoglycemic agent. 6) The length of follow-up varied between studies, from our cutoff of 12 weeks to as long as 208 weeks.

## Conclusions

In summary, this analysis suggests that SGLT2is may reduce the occurrence of AKI in patients with diabetes mellitus. This observation was seen in RCTs and observational cohort studies. There was no heterogeneity between the different SGLT2is. We suggest that a randomized placebo-controlled study should be performed in diabetic patients undergoing a medical procedure with a high risk of AKI (e.g., cardiothoracic on-pump surgery) to demonstrate the beneficial effects of SGLT2 inhibition on mechanisms of AKI.

## Supporting information

**S1 Text. PRISMA checklist.** PRISMA, Preferred Reporting Items for Systematic Reviews and Meta-Analyses.
(DOC)

**S2 Text. Analysis plan.**
(DOCX)

**S1 Table. Search strategy.**
(XLSX)

**S2 Table. Basic information about study type and AE definitions in the different RCTs.** AE, adverse event; RCT, randomized controlled trial.
(XLSX)

**S3 Table. Basic information about study type and AE definitions in the different observational cohorts.** AE, adverse event.
(XLSX)

**S4 Table. Number of patients treated with RASis in trials reporting AKI events.** AKI, acute kidney injury; RASi, renin–angiotensin system inhibitor.
(XLSX)

**S1 Fig. Risk-of-bias assessment in 41 studies reporting at least one AKI as SAE or AE.** AE, adverse event; AKI, acute kidney injury; SAE, serious AE.
(TIF)

**S2 Fig. Funnel plot.**
(TIF)

**S3 Fig. Effect of SGLT2is on serious AKI AEs after removal of one of the 28 studies reporting an SAE.** This analysis suggests that the overall result was not due to a single study. Even when the results of the largest study (DECLARE) was removed, the result remained highly significant ($p < 0.001$). AE, adverse event; AKI, acute kidney injury; SAE, serious AE; SGLT2i, sodium-glucose cotransporter-2 inhibitor.
(TIF)

**S4 Fig. Effect of SGLT2is on any AKI in RCTs.** AKI, acute kidney injury; RCT, randomized controlled trial; SGLT2i, sodium-glucose cotransporter-2 inhibitor.
(TIF)

**S5 Fig. Effect of SGLT2i on SAE AKI AEs in patients with eGFR <60 ml/min.** AE, adverse event; AKI, acute kidney injury; eGFR, estimated Glomerular Filtration Rate; SAE, serious AE; SGLT2i, sodium-glucose cotransporter-2 inhibitor.
(TIF)

**S6 Fig. Effect of SGLT2is on combined renal AEs in RCTs.** AE, adverse event; RCT, randomized controlled trial; SGLT2i, sodium-glucose cotransporter-2 inhibitor.
(TIF)

**S7 Fig. Effect of SGLT2is on hypovolemia-related AEs in RCTs.** (a) Canagliflozin, (b) dapagliflozin, (c) empagliflozin, (d) ertugliflozin, (e) other SGLT2is, and (f) comparison of estimates of all examined drugs. AE, adverse event; RCT, randomized controlled trial; SGLT2i, sodium-glucose cotransporter-2 inhibitor.
(TIF)

**S8 Fig. Effect of SGLT2is on AKI in propensity-score–matched observational cohorts.** AKI, acute kidney injury; SGLT2i, sodium-glucose cotransporter-2 inhibitor.
(TIF)

## Author Contributions

**Conceptualization:** Jan Menne.

**Data curation:** Jan Menne, Eva Dumann.

**Formal analysis:** Bernhard M. W. Schmidt.

**Methodology:** Bernhard M. W. Schmidt.

**Validation:** Jan Menne, Eva Dumann, Hermann Haller.

**Writing – original draft:** Jan Menne.

**Writing – review & editing:** Eva Dumann, Hermann Haller, Bernhard M. W. Schmidt.

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
