## [Decision Letter · Decision Letter 0]

20 Sep 2019

Dear Dr. Menne,

Thank you very much for submitting your manuscript "Effect of SGLT-2 inhibition on acute kidney injury: A systematic review and meta-analysis" (PMEDICINE-D-19-02279) for consideration at PLOS Medicine. 

[LINK]

In light of these reviews, I am afraid that we will not be able to accept the manuscript for publication in the journal in its current form, but we would like to consider a revised version that addresses the reviewers' and editors' comments. Obviously we cannot make any decision about publication until we have seen the revised manuscript and your response, and we plan to seek re-review by one or more of the reviewers. 

We expect to receive your revised manuscript by Oct 11 2019 11:59PM. Please email us (plosmedicine@plos.org) if you have any questions or concerns.

We look forward to receiving your revised manuscript. 

Sincerely,

Clare Stone, PhD

Managing Editor 

PLOS Medicine

plosmedicine.org

Abstract – please avoid the list like appearance and sectioning. The abstract should have 3 sections: Background, Methods and Findings and Conclusions. The final sentence of the Methods and Findings should be the limitations of the study. Please look at other published systematic reviews to see formatting for an abstract. 

Please use square brackets in the main text for references numbers instead of superscript.

Study Selection (Two independent investigators reviewed all articles) – please give initials of these authors. And then those for the 3rd. 

Remove “Role of the Funding Source: No external funding.” From the main text

I note some use of causal language (eg from abstract “SGLT2i prevents AKI with”). Even though this SR does feature trials there is also the inclusion of observational studies. Please avoid overheated language. 

Page 6 – “that there is indeed some tubular alteration” may instead of is.

Insert a subheading ‘Conclusions’ page 7

Remove conflict of interest from the main text

Comments from the reviewers:

Reviewer #1: Nice review, up-to-date on methods. Data are in line with results from the major large CV and renal outcome studies.

- Suggest to edit figures because the point estimated of the smaller studies are not visible.

- suggest to analyse a subgroup of studies that were double-blind 

- suggest to explain in the discussion that most randomised patients in the large trials received inhibitor of the RAS. Thus post-glomerular dilation (RASi) combined with pre-glomerular constriction (SGLT2i) should increase risk of AKI. The opposite was found and there is animal studies of experimental ischemic AKI showing much the same. 

Reviewer #2: Alex McConnachie

Menne et al report a meta-analysis of RCTs and observational studies of the effect of SGLT-2 inhibitors on AKI. This review looks at the use of statistics in the paper.

I have no major concerns with the analyses presented in the paper, just a few minor observations.

First, a typo: the final paragraph refers to the use of a "Funnel blot".

Then, an issue with wording: the second paragraph of the results states that "SGLT2i diminished the odds ratio by 32%"; this should perhaps be "SGLT2i diminished the odds by 32%".

There are a few places in the paper where the authors make statements about the effects of different SGLT-2 inhibitors being similar (e.g. in the section headed "AEs of AKI in RCT") or different (under "Hypovolemia related AE"), though they do not present any statistical tests to back up these statements. For hypovolemia, in particular, some sort of interaction test would be useful.

When talking about AKI in observational studies, there is a little confusion as to whether there are four studies, or five.

In the first paragraph of the discussion, it is stated that SLGT-2i reduce the incidence of acute renal failure. Is "incidence" correct here, or should "odds" be used?

Given that the included studies will have different lengths of follow-up, to what extent are the odds ratios directly comparable between studies? This could perhaps be listed among the limitations of the analysis.

The figures are generally good, though S8 is extremely hard to read. Would it be better to break it up into a number of figures, perhaps one for each individual drug, then give one showing how the drug-specific estimates combine to give an overall SGLT-2i effect estimate?

Reviewer #3: Well written, timely and clinically important. Although the emphasis was on AKI, I think it is also important to include all genitourinary AEs such as UTIs in this report. 

Reviewer #4: In the systematic review, the authors have included randomized control trials as well as observational studies of an SGLT2 inhibitor compared to controls. The outcomes were acute kidney injury, as defined on the basis of serious adverse event or adverse event reporting for the trials. In addition they also report hypovolemia similarly defined from the trials. Lastly they also have observational studies which reported acute kidney injury events.

Overall the authors report that the risk of acute kidney injury is decreased by about 24 to 32%, depending on which definition is used, and the trials and by 60% in the observational studies. In contrast, hypovolemia is increased by about 20%.

Overall the results do seem valid, however, this reviewer has some concerns about the interpretation, especially given the high heterogeneity, with respect to the difference between agents. The patient population and the studies does vary quite a lot, with the population in CREDENCE being at higher risk than in the other large trials. This may cast canagliflozin and an unfavorable light in some analysis and in favorable light in some other analysis. It is unlikely that these are true differences, versus a difference in baseline the risk and these conclusions should be drawn very cautiously.

The other major concern is that most of the data in this systematic review comes from the for major trials, CANVAS, EMPAREG, CREDENCE and DECLARE. The remaining trials add a very few numbers in the overall analysis. Unfortunately for the authors, a competing systematic review has recently been published which also reports a similar findings on acute kidney injury (25% RRR)from these 4 trials combined together https://www.thelancet.com/journals/landia/article/PIIS2213-8587(19)30256-6/fulltext. The difference of the present manuscript, is that they include different definitions of acute kidney injury, observational studies, and the outcome of hypovolemia. Separating the trial outcomes [ AKI and hypovolemia] and the observational studies [ acute kidney injury] would be helpful in the flow of the manuscript to make this apparent. Though this was not prespecified, some more analyses could help this paper stand out even more. For example, in the analysis where high heterogeneity was reported, some exploration of the heterogeneity would be useful. This could be done either a subgroup analysis or as Metaregressions. Understandably, these would all be post hoc analysis, however they may shed some more light on the underlying statistical heterogeneity.

Otherwise the discussion is clearly written, and emphasizes the lack of a mechanistic explanation, as well as the next steps.

[LINK]

---

## [Decision Letter · Decision Letter 1]

23 Oct 2019

Dear Dr. Menne,

Thank you very much for re-submitting your manuscript "Effect of SGLT-2 inhibition on acute kidney injury: A systematic review and meta-analysis" (PMEDICINE-D-19-02279R1) for review by PLOS Medicine.

I have discussed the paper with my colleagues and the academic editor and it was also seen again by reviewers. I am pleased to say that provided the remaining editorial and production issues are dealt with we are planning to accept the paper for publication in the journal.

[LINK]

We look forward to receiving the revised manuscript by Oct 30 2019 11:59PM. 

Sincerely,

Clare Stone, PhD

Managing Editor 

PLOS Medicine

plosmedicine.org

Requests from Editors:

- I think the authors need to say what SGLT-2 therapy is used for at the start of the abstract!

- Restructure abstract – the Methods and findings should be combined into one section, per house style. And please start the sentence on Limitations with “Limitations of this study are…” (instead of in terms of….)

- In the abstract, you say that SGLT-2 inhibitors had a "protective effect" in observational studies, which seems to overstate what can be shown from an observational study; I suggest "in the observational studies, the risk of adverse events was reduced in patients receiving SGLT2i" or similar

Remove ‘non-technical’ from the heading for the Author Summary

- Amend author summary to pull these 2 bullet points together, e.g. "Among 112 trials, 41 reported ..."

Did your study have a prospective protocol or analysis plan? Please state this (either way) early in the Methods section.

c) In either case, changes in the analysis—including those made in response to peer review comments—should be identified as such in the Methods section of the paper, with rationale.

- In the first paragraph of the discussion, "systemic"-> "systematic"

- square brackets in the wrong place for references. They should be before the full stop at the end of the sentence, not after. 

- Please provide a PRISMA checklist as a Supp file and please ensure that sections and paragraphs are used rather than page numbers as these can change on publication. 

Discussion – overstating findings : “As this systemic review is based on a high AKI event rate and we observed no heterogeneity, we conclude that SGLT2i may not only reduce the progression of chronic kidney disease, but also have a preventive effect on acute kidney injury.” A systematic review cannot show a preventative effect. Please remove. 

As above “However, in our analysis, we could find no evidence of this, further underscoring that reversible changes in glomerular haemodynamics caused by SGLT2i are probably of little relevance to hard outcomes such as AKI SAE and AEs.” Evidence cannot be shown in a systematic review – please remove. 

Conclusion “This observation is robust” – please remove the entire sentence – it’s vague and not quantified. 

Comments from Reviewers:

Reviewer #4: all previous comments addressed

Inclusion of DAPA-HF and other recent studies an additional strength

[LINK]

---

## [Editor Report · Decision Letter 2]

11 Nov 2019

Dear Dr. Menne, 

On behalf of my colleagues and the academic editor, Dr. Maarten Taal, I am delighted to inform you that your manuscript entitled "Acute kidney injury and adverse renal events in patients receiving SGLT2-Inhibitors: A systematic review and meta-analysis" (PMEDICINE-D-19-02279R2) has been accepted for publication in PLOS Medicine. 

PRODUCTION PROCESS

PRESS

PROFILE INFORMATION

Thank you again for submitting the manuscript to PLOS Medicine. We look forward to publishing it. 

Best wishes, 

Clare Stone, PhD

Managing Editor 

PLOS Medicine

plosmedicine.org